# Artemisinin Stimulates Neuronal Cell Viability and Possess a Neuroprotective Effect In Vitro

**DOI:** 10.3390/molecules30010198

**Published:** 2025-01-06

**Authors:** Sergey A. Pukhov, Alexey V. Semakov, Nadezhda E. Pukaeva, Olga A. Kukharskaya, Tatyana V. Ivanova, Viktoriya S. Kryshkova, Sergey O. Bachurin, Michail S. Kukharsky

**Affiliations:** 1Institute of Physiologically Active Compounds, Federal Research Center of Problems of Chemical Physics and Medicinal Chemistry, Russian Academy of Sciences, 142432 Chernogolovka, Russia; lnc@ipac.ac.ru (S.A.P.); l_vok@list.ru (A.V.S.); nadya.pukaeva98@gmail.com (N.E.P.); loa.ipac@yandex.ru (O.A.K.); vskryshkova@mail.ru (V.S.K.); kukharskym@gmail.com (M.S.K.); 2Department of General and Cell Biology, Faculty of Medical Biology, Pirogov Russian National Research Medical University, 117997 Moscow, Russia; vnv.ttn.v@gmail.com

**Keywords:** artemisinin, sesquiterpene lactone, neuroprotection, neurodegeneration, ER stress, aggregation

## Abstract

Artemisinin is a sesquiterpene lactone derived from the plant *Artemisia annua* L., renowned for its antimalarial activity. Based on this compound, various derivatives and analogues have been obtained that exhibit diverse biological activities, including clinically approved drugs. Recently, increasing evidence has highlighted the neuroprotective potential of artemisinin. In this study, we evaluated the effects of artemisinin on the viability of neuronal-like cells, including primary hippocampal neuronal cultures. Artemisinin exhibited a stimulating effect on SH-SY5Y and HEK-293 cells and enhanced the survival of primary neurons at low concentrations (1 µM). In contrast, artemisinin derivatives, such as dihydroartemisinin, anhydrodihydroartemisinin, and artemisitene, did not display similar stimulatory activity, suggesting that the intact lactone ring is crucial for this property. Furthermore, artemisinin demonstrated a protective effect against endoplasmic reticulum (ER) stress induced by the proteasome inhibitor MG132 in SH-SY5Y cells. However, it did not exhibit protective activity against oxidative stress induced by sodium arsenite. Additionally, artemisinin effectively inhibited the aggregation of mutated TDP-43 protein in transfected SH-SY5Y cells. These findings suggest that artemisinin exerts neuroprotective effects by targeting key molecular pathways associated with neurodegeneration, offering potential therapeutic insights for related conditions.

## 1. Introduction

Artemisinin (ART) is a sesquiterpene lactone of the cadinane structural type, characterized by its unique endoperoxide moiety, which sets it apart from other compounds in this class. This natural substance, a secondary metabolite derived from *Artemisia annua* L. (Asteraceae), was first recognized for its potent activity against the causative agent of tropical malaria (*Plasmodium* sp.) [1,2]. This discovery spurred the extensive development and application of ART-based derivatives for the treatment of malaria. While its antimalarial activity has been extensively studied and continues to be a topic of scientific discussion, ART’s broader therapeutic potential is garnering growing interest. Beyond its well-documented antimalarial properties, emerging research highlights its antitumor, anti-inflammatory, antidiabetic, and other beneficial effects [3,4]. Notably, recent studies have pointed to the neuroprotective potential of ART, opening new avenues for the investigation of its therapeutic potential [5,6,7].

The broad biological activity, well-characterized toxicity profile, and high availability of ART as a starting molecule for modification (the world production of ART is several hundred tons per year) suggest promising prospects for the development of ART-like molecules as potential candidates in medicinal chemistry [8,9]. In addition, ART can easily cross the blood–brain barrier that makes it a feasible candidate for the treatment of neurological diseases [10,11].

The neuroprotective effects of ART have been demonstrated in various neuron-like cell types. ART has been shown to suppress sodium nitroprusside (NO donor)-induced cell death in PC12 cells and primary cortical neurons [12]. Further studies revealed its protective effect against 6-OHDA (6-hydroxydopamine), MPP+ (1-methyl-4-phenylpyridinium ion), and β-amyloid toxicity in PC12 cells, where the effect was accompanied by activation of the extracellular signal-regulated kinase (ERK) pathway [5,13]. Additionally, ART prevented glutamate-induced oxidative injury in the HT-22 mouse hippocampal cell line by activating the Akt signaling pathway [14]. In experiments using SH-SY5Y cells (and primary cultures), ART protected against oxidative stress induced by hydrogen peroxide (H_2_O_2_) via activation of the adenosine monophosphate-activated protein kinase (AMPK) pathway [6]. ART also reduced apoptosis in SH-SY5Y cells following MPP+ treatment [15].

The neuroprotective properties of ART have also been observed in various in vivo models of neurodegenerative diseases (NDDs). These include 6-OHDA- and MPTP-induced Parkinson’s disease (PD) models in mice [13,16], transgenic Alzheimer’s disease (AD) mouse models [17,18,19], and a traumatic optic neuropathy model [7].

A key aspect of neurodegeneration involves cellular pathological events such as protein aggregation and endoplasmic reticulum (ER) stress [20,21,22]. While ER stress initially serves to balance protein production and degradation, chronic ER stress contributes to neurodegeneration by activating cell death pathways, leading to neuronal loss [23]. Developing therapeutic strategies targeting these detrimental processes is an attractive approach for combating NDDs.

In this study, we demonstrated that ART exhibits neuroprotective potential in neuronal cell lines, including primary neuronal hippocampal cultures. Specifically, ART increased the viability of neuroblastoma cells at micromolar concentrations and protected them from ER stress induced by the proteasome inhibitor MG132. Furthermore, ART inhibited the aggregation of TDP-43 protein, highlighting its effectiveness against specific neurodegeneration-related pathologies. Several ART derivatives, including its active metabolite dihydroartemisinin (DHA), were also evaluated as ART-like compounds. However, these derivatives showed higher toxicity in vitro compared to ART, and did not have a stimulating effect on cells, i.e., the change in the lactone functional group in ART leads to a loss of effect.

## 2. Results

### 2.1. Preparation of Artemisinin Analogues with Modifications in the Lactone Cycle

Based on the literature data, a sequence of transformations leading to artemisitene (AMT), a lactone with an activated exomethylene group, was proposed (Figure 1). It includes four stages. In the first stage, DHA can be obtained from ART by the known reduction reaction [24]. To reduce the number of stages, a commercially available DHA was chosen as the starting compound. When using boron trifluoride etherate as a dehydrating agent, anhydrodihydroartemisinin (ahDHA) is formed in quantitative yield [25].

The next stage includes the following two transformations: introduction of an oxygen atom into the site of the future lactone cycle of AMT and simultaneous formation of an exomethylene group. Under photooxygenation conditions, when interacting with singlet oxygen, the main reaction product (hydroperoxide) is obtained rather slowly, even when using a powerful light source. This may be caused by the use of air as a reactant instead of pure oxygen, and by the suboptimal selection of the solvent and photosensitizer combination for reaction. We assume that the modification of reaction conditions, by replacing the photosensitizer (in photooxygenation) or the singlet oxygen source (in catalytic decomposition of hydrogen peroxide), allows one to significantly reduce the reaction time, while achieving high yields of hydroperoxide. The resulting hydroperoxide is then converted to AMT with a high yield under the action of acetic anhydride and pyridine [26,27].

### 2.2. Artemisinin Increases Viability in SH-SY5Y and HEK-293 Cells, but Not in MCF-7 Cells

To examine the impact of ART on the survival of neuronal-like cells, we conducted an MTS assay on SH-SY5Y cells across a micromolar concentration range (0.1 to 500 µM). Notably, at 1 µM, ART significantly enhanced SH-SY5Y cell viability to 120% compared to control cells after a 48-h incubation period (Figure 2). Additionally, a trend towards increased viability was observed at 10 µM, though this was not statistically significant. Conversely, concentrations at or above 50 µM led to reduced cell viability, with an IC50 of 180 µM. Previous studies on ART’s cytotoxicity, including assessments on SH-SY5Y cells, did not report enhanced viability at low doses [6,15]. In contrast to these studies, the ART treatment time in our study was extended to 48 h instead of 24 h. Indeed, no effect was observed after 24 h of ART treatment. The IC50 for the 24-h time point was determined as 237 µM. This extended exposure may be key to understanding ART’s enhanced viability effect in neuronal-like cells at lower concentrations.

To evaluate whether the observed viability increase with ART is specific to SH-SY5Y cells, we also assessed its effect on HEK-293 and MCF-7 cells (Figure 2). The IC50 for HEK-293 cells was determined as 242 µM after 24 h and 232 µM after 48 h. A notable increase in HEK-293 cell viability was observed following 24-h treatment at concentrations from 1 to 100 µM, with the peak effect at 1 µM (117% of control). However, after 48 h, ART no longer had a significant impact on HEK-293 viability. This difference in the time window for stimulatory effects between HEK-293 and SH-SY5Y cells may reflect the shorter division time of HEK-293 cells compared to the slower-dividing SH-SY5Y cells [28,29]. Interestingly, no viability increase was observed in MCF-7 cells, regardless of treatment duration. This suggests that ART’s stimulatory effect is cell type-specific, enhancing viability in certain cell lines while showing no impact on others.

### 2.3. Artemisinin Does Not Affect the Cell Cycle, but Reduces ATP Levels

Next, we assessed the impact of ART analogues on SH-SY5Y cell viability. DHA is a metabolic derivative of ART and is considered the active substance responsible for its antimalarial effects [30,31]. AMT and ahDHA are semisynthetic derivatives of artemisinin that contain modifications in the lactone cycle but retain the unaltered endoperoxide 1,2,4-trioxane ring [26]. These derivatives were included in the analysis to evaluate the importance of the lactone cycle moiety for the activity of the molecules. AMT and DHA exhibited high toxicity, with IC50 values of 5 μM and 0.9 μM, respectively (Figure 3A), while ahDHA showed moderate toxicity, with an IC50 of 83 μM (See Appendix A). None of the tested compounds demonstrated a stimulatory effect at any concentration. These results indicate that the observed effects of ART are closely related to the structure–activity relationship.

Assuming that ART may affect cell proliferation, we assessed the distribution of SH-SY5Y cells among cell cycle phases using flow cytometry. The measurement of DNA content allows for studying cell populations in different phases of the cell cycle, as well as DNA ploidy analysis. Cells in a given population are typically distributed among the following three main phases: the G0/G1 phase (characterized by one set of paired chromosomes per cell), the S phase (where DNA synthesis occurs, resulting in variable DNA content), and the G2/M phase (characterized by two sets of paired chromosomes per cell prior to division). During the 48-h treatment with the selected compounds (1 μM), none of them led to significant changes relative to the control. Histograms of SH-SY5Y cells stained with FxCycle™ dye, showing the distribution of DNA content, are presented in Figure 3B.

The total cellular ATP level can be used as an interpretation of the biological response when assessing cell viability and proliferation, as well as the cytotoxicity of compounds. We assessed the impact of various concentrations of the compounds on ATP levels in SH-SY5Y cells after a 48-h treatment. ART caused a slight but consistent reduction in ATP levels at concentrations ranging from 1 to 200 μM. A comparable trend was observed for DHA, while AMT demonstrated a markedly stronger effect, nearly abolishing the cellular energy balance at concentrations starting at 10 μM (Figure 3C).

### 2.4. Artemisinin Increases Survival and Impact Branching of Neurons in Primary Hippocampal Culture

As a more relevant model for the nervous cells, we utilized a mixed neuron–glia murine hippocampal primary culture (Figure 4A). ART was added to the culture media from the first day of cultivation, with assessments on day 7 in vitro (DIV). In this primary culture, ART showed greater toxicity compared to SH-SY5Y cells, with a significant reduction in cell viability beginning at 10 µM, while 1 µM exhibited no toxic effect (Figure 4B). Notably, we did not observe any stimulatory effects at any concentration.

We also assessed neuron numbers and neurite arborization in primary culture after 7 days of treatment with 1 µM ART, using Sholl analysis. The standard medium for the primary neuronal culture included B27 supplement, which is known to support neuron survival in culture [32]. Omitting B27 from the medium resulted in a substantial decline in neuron numbers and reduced branching (Figure 4C–E). Under these suboptimal conditions, a trend (though not statistically significant) towards an increased number of neurons (identified using the NeuN marker) and significantly elevated neurite branching was observed in ART-treated cultures. In optimal media containing B27, ART further increased the percentage of neurons in the culture; however, it led to a decrease in neurite branching compared to cultures with B27 alone (Figure 4C–E).

### 2.5. Artemisinin Protects SH-SY5Y Cells from Stress Induced by the Proteasome Inhibitor MG132

We investigated the potential protective effects of ART against the following two types of cellular stresses associated with NDDs: ER stress induced by the proteasome inhibitor MG132 and oxidative stress induced by sodium arsenite (SA). SH-SY5Y cells were treated with ART at concentrations of 1, 10, and 50 µM for 48 h. During the last 24 h of incubation, 1 µM MG132 or 50 µM SA were added to the media. These stress-inducing treatments reduced the cell viability to approximately 80% and 60% of the control group, respectively (Figure 5A,B).

ART at 1 µM effectively protected cells from MG132-induced stress, restoring the viability to levels comparable to the control group (Figure 5A). Higher concentrations (10 and 50 µM) did not show significant protective effects compared to stressed cells without ART treatment. Notably, ART did not impact cell viability under oxidative stress induced by SA (Figure 5B), suggesting a specific protective effect of ART against MG132-induced stress.

### 2.6. Artemisinin Inhibits Aggregation of Mutated TDP-43 in SH-SY5Y Cells

ER stress in NDDs is commonly triggered by the accumulation of protein aggregates in neuronal cells. We investigated whether ART could inhibit such protein aggregation. SH-SY5Y cells were transfected with a plasmid encoding the truncated form of TDP-43(Δ1–192), a protein highly prone to aggregation [33], and fused with GFP. In a subset of transfected cells, GFP-TDP-43(Δ1–192) formed medium to large, punctate-like aggregates dispersed throughout the cytoplasm (Figure 6A upper panel). Cell cultures were treated with 1 µM (Figure 6A middle panel) or 10 µM (Figure 6A lower panel) ART 24 h after transfection, and the number of cells with aggregates was subsequently quantified. ART treatment significantly reduced TDP-43 aggregation at both concentrations (Figure 6B). Importantly, the total number of cells, as estimated by nuclear DAPI staining, did not differ between the treated and untreated cultures, indicating comparable levels of cell death under both conditions. These findings demonstrate that ART is not only protective against ER stress, but also reduces the misfolded protein burden contributing to ER stress in NDDs contexts.

## 3. Discussion

We demonstrated that artemisinin (ART) stimulates viability in the neuronal SH-SY5Y cell line at 1 µM, but not in the epithelial-origin MCF-7 cells. Interestingly, ART also stimulated HEK-293 cells, which, though derived from human embryonic kidney cells, are known to express neuronal genes, and are believed to have an embryonic neural cell origin [34,35,36]. This suggests that ART’s stimulatory effect may be specific to neuronal cell types.

The closest structural derivatives of ART are dihydroartemisinin (DHA) and artemisitene (AMT), and despite their similar effects of action with the original molecule in other types of activity, they exhibit a high level of cytotoxicity on SH-SY5Y cells (IC50, at about 1 and 10 µM, respectively), and we did not find a stimulating effect in the selected concentration range. Thus, the effect of ART is probably not associated with a direct effect on cell proliferation, since, at a stimulating concentration, it does not affect the cell cycle. ART reduced the ATP level in SH-SY5Y cells at concentrations of 1 to 200 µM after 48 h of incubation, but increased it at concentration of 0.1 µM. It can be speculated that lower concentrations or shorter incubation times with ART may stimulate ATP production, whereas higher concentrations or prolonged incubation periods may have the opposite effect. Further investigation is required to elucidate the time-dependent impact of ART on cellular energy balance, particularly by analyzing ATP levels at different time points or using a real-time detection format. Although DHA is a known active metabolite of ART, its high cytotoxicity led us to focus exclusively on ART in subsequent analyses.

Further, we investigated ART’s effects on primary neuronal culture. Because primary neurons are non-dividing, unlike immortalized cell lines such as SH-SY5Y, we adopted a different treatment strategy. Cells were incubated with a non-toxic concentration of ART (1 µM) during the first week of cultivation, when neural precursor cells survive and differentiate. The number of differentiated neurons (marked by NeuN) was approximately 1% in non-optimal conditions without the B27 supplement, and about 3% in cultures without B27 but treated with ART. However, this difference did not reach statistical significance. In optimal conditions with B27, ART raised the proportion of NeuN-positive cells to 31%, compared to 16% in B27-only media. Interestingly, ART also had an ambivalent effect on neurite outgrowth, as it promoted branching in B27-lacking conditions, but reduced it when B27 was present. Further research is needed to understand how these condition-specific effects may influence neuronal activity. This experiment indicates that prolonged exposure to low-dose ART may enhance neuronal survival and confer protective effects.

Neuroprotection is a key therapeutic strategy for NDDs, where neuronal cell damage and death are prevalent [37,38]. To further examine ART’s neuroprotective potential, we investigated its effects on cellular pathologies associated with neurodegeneration. Extracts of *Artemisia annua* L. and its main component, ART, are known for their antioxidant properties [39,40,41]. ART has also been shown to protect SH-SY5Y cells from oxidative stress induced by H_2_O_2_ [6]. In this work, the authors do not note a stimulatory effect of ART on the SH-SY5Y cell line (the authors were limited to only a 24-h experiment). However, it was shown that ART attenuated the decrease in cell viability induced by H_2_O_2_ in SH-SY5Y cells (pretreatment for 2 h, exposure to H_2_O_2_ for 24 h). Additionally, in this study, we found that ART did not protect against oxidative stress induced by SA (pretreatment for 24 h, exposure to SA for additional 24 h), another inducer of oxidative damage. This discrepancy may arise from differences in the mechanisms and signaling pathways activated by H_2_O_2_ and SA, as well as due to the different times of treatment [42].

We further demonstrated that ART at 1 µM, the same concentration that stimulated SH-SY5Y cells, protected them from ER stress induced by the proteasome inhibitor MG132. Interestingly, higher concentrations (10 and 50 µM) did not have this protective effect, although a slight non-significant increase in viability was observed at 10 µM. A biphasic, dose-dependent response has also been reported with ART analogs in other studies. For example, high doses of artemether and artesunate caused neurotoxicity or mortality in Swiss albino mice [43], and artesunate exhibited cardiotoxicity at high doses but conferred cardioprotective effects at low doses in zebrafish [44]. Similarly, we observed that high concentrations of ART (IC50 180 µM) were toxic to SH-SY5Y cells, whereas lower concentrations (1 µM) were beneficial.

The evidence suggests that DHA, the active form of ART, triggers an unfolded protein response (UPR) in the malaria-causing *Plasmodium falciparum*, elevating eIF2α phosphorylation and inhibiting proteasome function. This response underpins ART’s effectiveness in malaria treatment [45]. In ART-resistant strains, over-activation of the ubiquitin–proteasomal and autophagic pathways further supports the involvement of these stress response mechanisms in ART’s action [46,47,48]. It is plausible that low doses of ART or its analogs partially engage stress response mechanisms, thus reinforcing cellular resilience to more severe stress. This hypothesis may explain the cytoprotective effects observed in neuronal cells in our in vitro experiments, though further study is needed to fully elucidate this potential mechanism in neuroprotection.

Multiple studies indicate that ART and its analogs inhibit protein aggregation in mouse models of Alzheimer’s disease (AD), where they reduce amyloid deposits in the cortex and hippocampus, with a greater effect observed at lower doses (10 mg/kg compared to 100 mg/kg) [49,50,51]. Our findings extend this anti-aggregation potential to TDP-43, a protein implicated in NDDs such as amyotrophic lateral sclerosis and frontotemporal dementia [52,53]. In SH-SY5Y cells transfected with the pathogenic TDP-43(Δ1–192) variant, ART at 1 and 10 µM significantly reduced the proportion of cells with visible aggregates. This highlights ART’s broader potential to target pathogenic protein aggregation beyond amyloid in AD models.

## 4. Materials and Methods

### 4.1. Compounds

Artemisinin and dihydroartemisinin were obtained from commercial sources (Shanghai Huirui Chemical Technology, China, and Greenherb Biological Technology (Xi’an), China). ART derivatives—anhydrodihydroartemisinin and artemisitene—were prepared as described previously [26]. All compounds were consistent with those described in the literature.

### 4.2. Stable Cell Lines, Cell Viability Assay

HEK293 and MCF-7 cells were maintained in Dulbecco’s Modified Eagle Medium (DMEM, Paneco, Moscow, Russia), and SH-SY5Y cells were cultured in DMEM/F12 medium (Paneco, Moscow, Russia) supplemented with 10% fetal bovine serum (Biosera, Cholet, France) and 2 mM L-glutamine (Paneco, Moscow, Russia). All cells were incubated at 37 °C in a humidified 5% CO₂ atmosphere. Cell lines were obtained from the Russian Cell Culture Collection (Institute of Cytology, Russian Academy of Sciences, St. Petersburg, Russia).

Cell viability was assessed using the MTS Assay Kit (Abcam, Cambridge, UK) according to the manufacturer’s instructions. Cells were plated in 96-well plates at a density of 1 × 10⁴ cells in 100 µL of medium. After 24 h, testing compounds were added (the final concentrations were 500, 250, 125, 62.5, 2, 1, and 0.5 μM for ART and ahDHA; 20, 10, 5, 2.5, 1.25, 0.625, and 0.3125 for AMT and DHA), bringing the final volume per well to 200 µL. Cells were incubated for an additional 24 or 48 h under the same conditions. All compounds were dissolved in DMSO, ensuring a final concentration not exceeding 0.5% DMSO, which did not induce cytotoxic effects. Control wells received an equivalent concentration of DMSO. After incubation, 20 µL of MTS reagent was added to each well, and plates were incubated for 1 to 2 h. Optical density was measured at 490 nm using a Feyond-A300 Microplate Reader (Allsheng, Hangzhou, China). IC50 values were calculated based on dose-response curves using GraphPad Prism 6 software (GraphPad Software, San Diego, CA, USA). The cell viability in control wells was set to 100%.

The protective effect against stress was determined using the MTS assay, as mentioned above. Cells were seeded in a 96-well plate at 1 × 10^4^ cells/100 μL complete growth medium, and cultured at 37 °C under CO_2_ (5%). After 24 h of incubation, ART was added to the cells (final concentrations were 50, 10, and 1 μM) by replacing 50% of the medium with a final volume of 100 μL/well. After 24 h of cultivation with ART, MG132 and NaAsO_2_ (SA) solutions in the growth medium with a final content of 1 and 50 μM, respectively, were added to the cells, and the initial concentration of ART was maintained. For each concentration, the experiment included three replicates. All substances were dissolved in DMSO; the final DMSO content in the well did not exceed 0.5% and did not have a toxic effect. The solvent was added to the control wells with a volume of 0.5% (positive control, treatment with MG132 and NaAsO_2_; additional control, treated only with ART). After another 24 h of incubation, 20 μL of MTS reagent were added to each well, and the plates were further incubated for 1 to 2 h.

A plasmid vector encoding a mutant form of TDP-43 protein (Δ1–192) with a GFP tag was provided by Prof. V. Buchman. SH-SY5Y cells were transfected using Lipofectamine 2000 reagent (Thermo Fisher Scientific, Waltham, MA, USA), following the manufacturer’s instructions. Twenty-four hours post-transfection, cells were fixed and stained with DAPI (Sigma-Aldrich, Burlington, MA, USA) for nuclear visualization.

### 4.3. Cell Cycle Analysis

Flow cytometry was used to assess the effect of the test compounds on the cell cycle of SH-SY5Y cells. Cells were seeded in a 12-well plate (1 × 10^6^ cells in 2000 µL) and solutions of the test compounds were added (final concentrations was 1 μM), then incubated for 48 h. FxCycle™ Violet Stain (Thermo Fisher Scientific, Waltham, MA, USA) reagent was used in the experiment. Cell material was prepared according to the reagent manufacturer’s instructions, cells were fixed with cold ethanol (70%) for 15 min, and then they were washed with phosphate-buffered saline (PBS). Then, they were analyzed using an Attune NxT Acoustic Focusing Cytometer, using a 405 nm laser with a 440/50 bandpass filter to achieve 50,000 events at a standard flow rate of 100 µL/min.

### 4.4. ATP Level Assessment

SH-SY5Y cells were seeded in a 96-well plate at 2.5 × 10^4^ cells/100 μL of complete growth medium, and cultured at 37 °C in a CO_2_ atmosphere (5%). After 24 h of incubation, the medium was removed and the studied compounds were added to the cells (final concentrations were 200, 100, 10, 1, 0.5, and 0.1 μM), with a final volume of 100 μL/well; then, the cells were cultured under the same conditions for 48 h. For each concentration, the experiment included three replicates. All substances were dissolved in DMSO, and the final DMSO content in the well did not exceed 0.2%. The solvent was added to the control wells with a volume of 0.2%. After 48 h, the cell material was prepared according to the manufacturer’s instructions for the Luminescent ATP Detection Assay Kit (Abcam, Cambridge, UK), and the luminescence in the wells was determined using a Cytation3 imaging reader (BioTek, Winooski, VT, USA).

### 4.5. Primary Hippocampal Cultures

Primary neuronal cultures were prepared from wild-type C57Bl/6J mice on postnatal day 3 (P3), as previously described [54]. Animals were housed under standard conditions, with a 12-h light/dark cycle and free access to food and water. Following dissection, hippocampi were incubated in 0.1% trypsin in Hank’s Balanced Salt Solution (HBSS, Paneco, Moscow, Russia), containing 10 mM HEPES (4-(2-hydroxyethyl)-1-piperazineethanesulfonic acid, Paneco, Moscow, Russia) and 1 mM sodium pyruvate (Paneco, Moscow, Russia) for 40 min. Mechanical dissociation was performed in Neurobasal medium (Paneco, Moscow, Russia), supplemented with 50 U/mL penicillin-streptomycin, 0.2% β-mercaptoethanol, 500 μM L-glutamine, 0.36% glucose, and 10% horse serum (Paneco, Moscow, Russia).

Hippocampal cells were then centrifuged at 1500 rpm for 5 min, and the pellet was resuspended in freshly prepared medium. To support neuronal survival, a B27 supplement (Thermo Fisher Scientific, Waltham, MA, USA) was added according to the experiment design. Cells were plated onto 12-mm diameter poly-L-lysine-coated coverslips at a density of 3 × 10⁴ cells per coverslip. The following day, the medium was replaced with serum-free fresh medium, and half of the medium was subsequently replaced every three days. Cells were fixed and stained on day 7 in vitro. On the first day in vitro, ART dissolved in DMSO (Paneco, Moscow, Russia) was added to a final concentration of 1 μM, which was maintained throughout the culture period.

### 4.6. Immunocytochemical Staining

Cells were rinsed with PBS and fixed in 4% paraformaldehyde for 15 min, followed by permeabilization with cold methanol for 5 min. After washing with PBS and blocking in 5% goat serum in PBS with 0.1% Tween-20 for 60 min at room temperature, coverslips were incubated with primary antibodies against Tau (1:1000, SAB4300377, Sigma-Aldrich, Burlington, MA, USA) and NeuN (1:1000; MAB377, Millipore, Burlington, MA, USA) for 60 min at room temperature. Cells were then incubated with secondary antibodies, Goat anti-Rabbit IgG Alexa Fluor™ 568 (A-11011, Thermo Fisher Scientific, Waltham, MA, USA) and Goat anti-Mouse IgG Alexa Fluor™ 488 (A-11029, Thermo Fisher Scientif, ic, Waltham, MA, USA), in PBS with 0.4% Tween-20 for 90 min at room temperature. Nuclei were stained with DAPI (Sigma-Aldrich, Burlington, MA, USA). Coverslips were mounted onto slides with Immu-Mount medium (Thermo Fisher Scientific, Waltham, MA, USA). Stained coverslips were imaged using the Cytation3 imaging reader and Gen5 3.08 software (BioTek, Winooski, VT, USA). An area of 3000 × 3000 µm was scanned in multi-channel fluorescence mode, stitched into a single panoramic image, and analyzed for cell counts stained with specific markers. Results for each marker were normalized to the total cell count, as determined by DAPI-stained nuclei. To assess neurite branching in primary neurons, microphotographs were captured with a Carl Zeiss Axio Observer 3 microscope equipped with an Axiocam 712 mono camera (Carl Zeiss, Oberkochen, Germany). Semi-automated Sholl analysis was conducted in ImageJ 1.52n, as previously described [55], with a total of 60 neurons analyzed per group.

### 4.7. Statistical Analysis

Statistical data analysis was performed using GraphPad Prism 6 software (GraphPad Software, San Diego, CA, USA). In all cases, the results are presented as the mean ± standard error. Details of the statistical analysis for each dataset are presented in the figure legends. Differences between groups were considered statistically significant at *p* < 0.05.

## Figures and Tables

**Figure 1 molecules-30-00198-f001:**
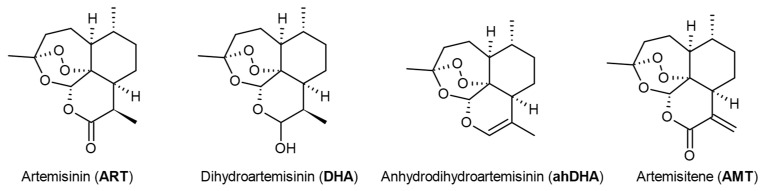
Structure of artemisinin and its derivatives.

**Figure 2 molecules-30-00198-f002:**
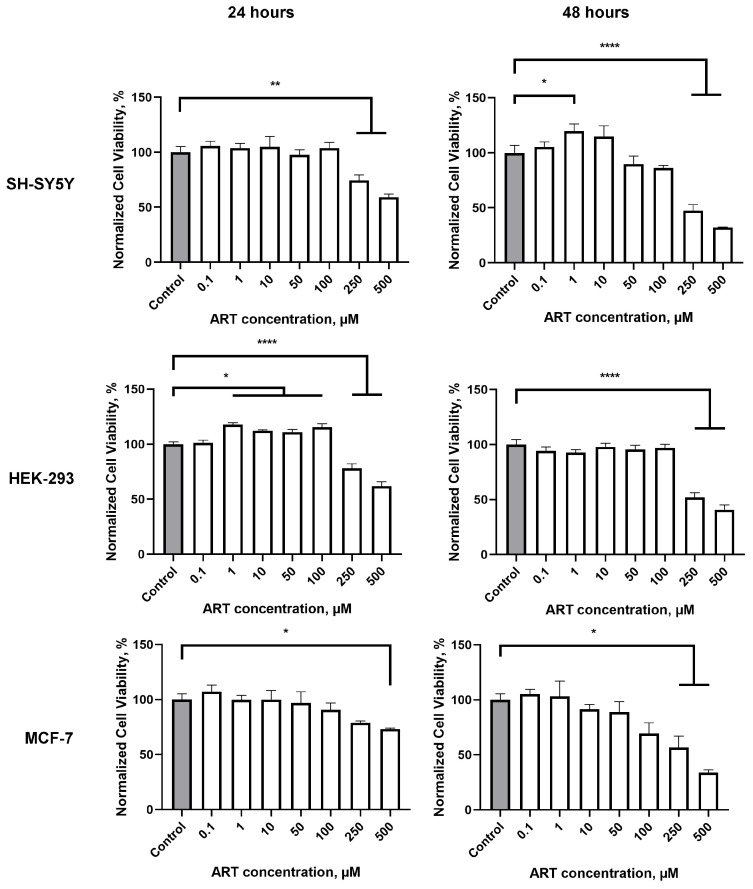
Viability of SH-SY5Y, HEK-293, and MCF-7 cells 24 h and 48 h after treatment with artemisinin (ART), across a range of micromolar concentrations measured using the MTS assay. Statistical analysis was performed using ANOVA, followed by Fisher’s LSD test for multiple comparisons between groups; * *p* < 0.05, ** *p* < 0.01, **** *p* < 0.0001 compared to control.

**Figure 3 molecules-30-00198-f003:**
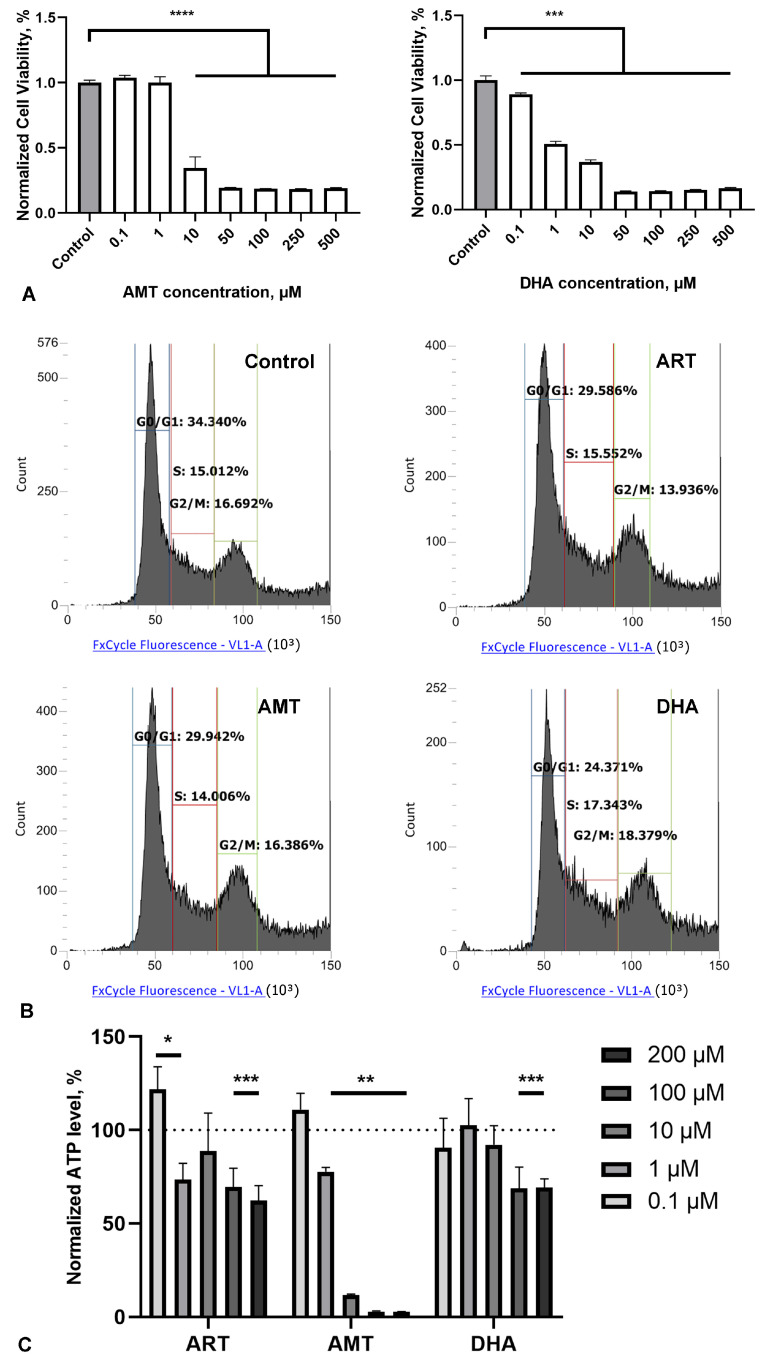
Effect of artemisinin (ART) and its derivatives—artemisitene (AMT) and dihydroartemisinin (DHA)—on SH-SY5Y cells’ viability, proliferation, and ATP level. (**A**) Viability of SH-SY5Y cells 48 h after treatment, with tested compounds across a range of micromolar concentrations measured using the MTS assay. Statistical analysis was performed using ANOVA, followed by Fisher’s LSD test for multiple comparisons between groups; *** *p* < 0.001, **** *p* < 0.0001 compared to control. (**B**) Representative histograms of untreated cells and 48 h after incubation with the tested compounds in concentration of 1 μM. (**C**) ATP level in SH-SY5Y cells relative to control 48 h after incubation, with the tested compounds in a concentration of 1 μM. Statistical analysis was performed using two-way ANOVA, followed by Fisher’s LSD test for multiple comparisons between groups; * *p* < 0.05, ** *p* < 0.01, *** *p* < 0.001 compared to control.

**Figure 4 molecules-30-00198-f004:**
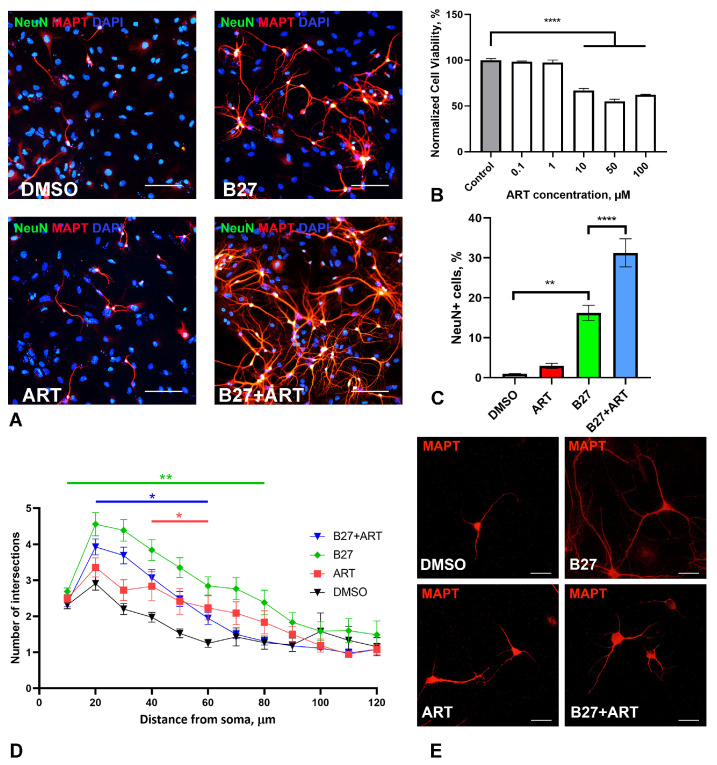
Effect of artemisinin (ART), B27 supplement, and their combination (B27+ART) on primary hippocampal culture. (**A**) Representative micrographs showing immunohistochemical staining of primary neurons (NeuN, a neuronal nuclei marker) and their neurites (MAPT, a microtubule-associated protein tau) in cultures incubated with DMSO (control), ART, B27, and B27+ART for first 7 days in vitro. Cell nuclei were stained using DAPI. The scale bar: 100 μm. (**B**) Viability of primary cultures treated with ART across a range of micromolar concentrations, measured using the MTS assay. (**C**) Quantification of neuronal number (NeuN-positive cells) under different incubation conditions (DMSO, ART, B27, and B27+ART) in the same cultures shown in panel (**A**). (**D**) Sholl analysis of primary neurons under different incubation conditions, showing the number of neurite intersections with 10-μm-spaced concentric shells as a function of the radial distance from the soma. (**E**) Representative micrographs of MAPT-stained primary neurons used for Sholl analysis. The scale bar: 20 μm. Statistical analysis was performed using ANOVA, followed by Fisher’s LSD test for multiple comparisons between groups; * *p* < 0.05, ** *p* < 0.01,**** *p* < 0.0001 compared to control.

**Figure 5 molecules-30-00198-f005:**
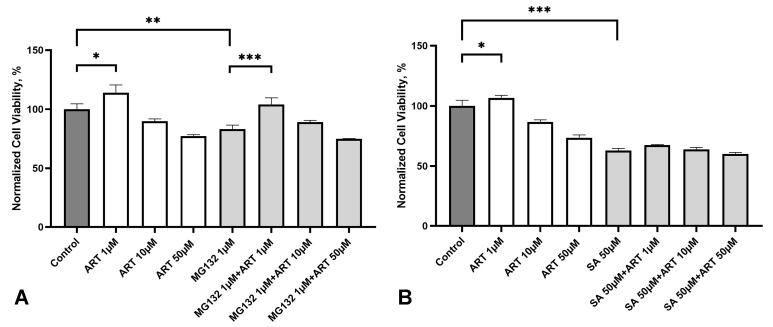
Effect of artemisinin (ART) on the viability of SH-SY5Y cells under stress conditions. (**A**) Viability of SH-SY5Y cells treated with the proteasome inhibitor MG132 (1 μM) and ART at the indicated concentrations, measured using the MTS assay. (**B**) Viability of SH-SY5Y cells exposed to oxidative stress induced by sodium arsenite (SA, 50 μM) and ART at the indicated concentrations, measured using the MTS assay. Statistical analysis was performed using ANOVA, followed by Fisher’s LSD test for multiple comparisons between groups; * *p* < 0.05, ** *p* < 0.01, *** *p* < 0.001 compared to control.

**Figure 6 molecules-30-00198-f006:**
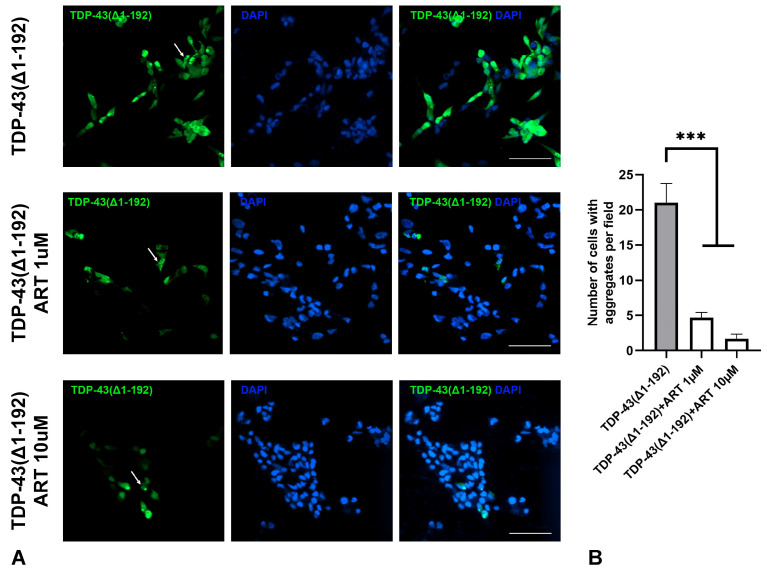
Effect of artemisinin (ART) on the aggregation of mutated TDP-43(Δ1–192) protein in SH-SY5Y cells. (**A**) Representative micrographs of SH-SY5Y cells transfected with GFP-tagged TDP-43(Δ1–192) protein and treated with ART at indicated concentrations. Cell nuclei were stained with DAPI. The scale bar: 50 μm. (**B**) Quantification of the number of cells with aggregates in SH-SY5Y cultures transfected with TDP-43(Δ1–192) and treated with ART. Statistical analysis was performed using ANOVA, followed by Fisher’s LSD test for multiple comparisons between groups; *** *p* < 0.001 compared to control group without ART treatment.

## Data Availability

Data are contained within the current article and its Appendix A.

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
