# Peer review of "Artemisinin Stimulates Neuronal Cell Viability and Possess a Neuroprotective Effect In Vitro"

_molecules, 2025, doi:10.3390/molecules30010198_

Round 1
Reviewer 1 Report
Comments and Suggestions for Authors
The objective of this study was to investigate the bioactive potential of Artemisinin, with a particular emphasis on evaluating its neuroprotective potential in primary cell cultures. The neuroprotective potential of Artemisinin has been extensively documented in the specialized literature, as noted in the abstract of the present study. However, certain aspects of the study would benefit from further clarification. Specifically, it is recommended that the abstract be reorganized to clearly articulate and highlight the novel contributions and unique findings of this research in order to enhance its academic impact.
Line 102 It is not specified in the methods section whether cellular viability was assessed for the neuroblastoma cell line. Please clarify this point and make the necessary corrections to ensure accuracy and completeness.
Additionally, could you elaborate on the rationale for selecting and comparing these specific cell lines?
Line 105: How do you explain a reported cellular viability of 120%? Please clarify this result, as values exceeding 100% may require additional context or justification, such as normalization methods or specific experimental conditions, to ensure accurate interpretation.
Line 125: Could you please elaborate on and provide clarification for this statement? A more detailed explanation would enhance the reader's understanding of its context and relevance within the study.
Line 134: Could you please explain why only a single dose and a single cell line were evaluated? Providing a rationale for this choice would help clarify the experimental design and its limitations.
Line 173: Could you please specify the method employed for the identification of primary cells?
Please specify the names of the products used in the study in lines 329–332 to ensure clarity and reproducibility.
Line 330: Were the ART derivatives prepared specifically for this study, or were they sourced from other studies?
Line 335: Why did you choose to use these two stabilized cell lines, specifically HEK293 and MCF-7? What was the rationale behind selecting one normal (embryonic renal) and one tumoral (breast cancer) cell line? Additionally, at what passage were the cell lines used in your experiments?
Were the tested concentrations derived from other studies, or were they established specifically for this study?
Line 369: The SH-SY5Y cell line was not mentioned in the initial description of the cell lines provided in line 335. Could you clarify its inclusion in the study?
Furthermore, please specify the source of the SH-SY5Y cell line, the conditions under which it was maintained, and the passage number at which it was used.
Lastly, indicate the precise quantity of plasmid employed for the transfection experiments to ensure reproducibility and clarity.
Line 374: Could you clarify why the cell cycle analysis by flow cytometry was not conducted for the other treated cell lines?
Additionally, the methodology detailing how the neuroblastoma SH-SY5Y cells were treated with the compounds has not been described (at the section line 334). Providing this information would enhance the comprehensiveness and reproducibility of the study. What was the rationale for selecting a final concentration of 1 µM for the compound? Could you elaborate on the reasoning behind this choice?
Additionally, why were all potential concentrations not tested? Providing clarification on this decision would strengthen the study's methodological transparency.
Line 368: Could you please clarify which cell lines were used for ATP evaluation? Providing this information would enhance the clarity and specificity of the methodology.
Line 398: Did you obtain approval from the ethics committee for initiating the primary culture? If so, please specify the approval number for transparency.
Line 408: How were the cells identified? Using the described method, were the cells isolated specifically from the hippocampus? Without cell sorting, it is likely that the culture remained heterogeneous after seven days of cultivation. Could you provide clarification on this aspect?
Why was only a single concentration selected for testing? Expanding on this decision would help in understanding the experimental design.
Line 416: What was the total cultivation period? Specifying this duration would enhance the methodological clarity of the study.
To enhance the clarity and rigor of the study, it is important to explicitly outline its limitations, particularly in terms of methodological constraints or potential sources of bias. Furthermore, a comparative analysis with similar studies in the existing literature is essential to highlight the novel aspects and unique contributions of this research. Finally, I encourage the inclusion of well-supported conclusions that not only summarize the findings but also provide meaningful insights into their broader implications and potential future directions for research.
Reviewer 2 Report
Comments and Suggestions for Authors
In this manuscript, Pukjov et al evaluated artemisinin's effects on neuronal-like cells, including primary hippocampal neurons. At low concentrations artemisinin stimulated SH-SY5Y and HEK-293 cell viability and enhanced primary neuron survival. Artemisinin protected SH-SY5Y cells against MG132-induced endoplasmic reticulum stress but not against sodium arsenite-induced oxidative stress induced. Further, Artemisinin inhibited the aggregation of mutated TDP43 protein highlighting its therapeutic potential for neurodegenerative disorders. The manuscript is comprehensive and well written. However, there a few concerns that need to be addressed.
1. There are a lot of references that are missing. For example-
a. Line 36-37- This natural substance, a secondary metabolite derived from Artemisia annua L. 35 (Asteraceae), was first recognized for its potent activity against the causative agent of 36 tropical malaria (Plasmodium sp.).
b. Line 110-111 -Previous studies on ART's cytotoxicity, including assessments on SH-SY5Y 110 cells, did not report enhanced viability at low doses.
2. The last paragraph of the introduction should be re-written and focus on what has been currently done in the study and the bigger implications of the study.
3. What was the rationale behind making ART analogues? This should be explained.
4. How many times were the experiments performed? This information should be included in each figure legend and in the methods section. All the graphs should show the individual data points. If 3 experiments were performed, the column graph should show all 3 data points (wherever applicable).
5. The author needs to discuss in great detail in the discussion section the correlation between decreased ATP levels and increased cell survival. This information is missing.
6. When the authors say neuron numbers, do they mean soma count? If so, I would recommend using "soma count" instead. Figure 4C, D and F lacks statistics. What scale bar was used for Fig 4E? Expand MAPT.
7. The authors show that 1μM ART does not affect cell viability in primary neurons. Interestingly, the same concentration increases in neuron number. How do they explain this? This issue needs to be addressed.
8. For Fig 6B. Include the statistical difference between TDP-43(Δ1–192)+ 1μM and TDP-43(Δ1–192)+ 10μM
Round 2
Reviewer 2 Report
Comments and Suggestions for Authors
The authors have addressed all my concerns.